# Identifying Soybean Pod Borer (*Leguminivora glycinivorella*) Resistance QTLs and the Mechanism of Induced Defense Using Linkage Mapping and RNA-Seq Analysis

**DOI:** 10.3390/ijms231810910

**Published:** 2022-09-18

**Authors:** Liangyu Chen, Baixing Song, Cheng Yu, Jun Zhang, Jian Zhang, Rui Bi, Xueying Li, Xiaobo Ren, Yanyu Zhu, Dan Yao, Yang Song, Songnan Yang, Rengui Zhao

**Affiliations:** 1Faculty of Agronomy, Jilin Agricultural University, Changchun 130118, China; 2National Crop Variety Approval and Characteristic Identification Station, Jilin Agricultural University, Changchun 130118, China; 3Department Biology, University of British Columbia-Okanagan, Kelowna, BC V1V 1V7, Canada; 4College of Plant Protection, Jilin Agricultural University, Changchun 130118, China; 5College of Life Science, Jilin Agricultural University, Changchun 130118, China

**Keywords:** biological stress, soybean pod borer resistance, linkage mapping, transcriptomics, soybean pod shells

## Abstract

The soybean pod borer (*Leguminivora glycinivorella*) (SPB) is a major cause of soybean (*Glycine max* L.) yield losses in northeast Asia, thus it is desirable to elucidate the resistance mechanisms involved in soybean response to the SPB. However, few studies have mapped SPB-resistant quantitative trait loci (QTLs) and deciphered the response mechanism in soybean. Here, we selected two soybean varieties, JY93 (SPB-resistant) and K6 (SPB-sensitive), to construct F_2_ and F_2:3_ populations for QTL mapping and collected pod shells before and after SPB larvae chewed on the two parents to perform RNA-Seq, which can identify stable QTLs and explore the response mechanism of soybean to the SPB. The results show that four QTLs underlying SPB damage to seeds were detected on chromosomes 4, 9, 13, and 15. Among them, *qESP-9-1* was scanned in all environments, hence it can be considered a stable QTL. All QTLs explained 0.79 to 6.09% of the phenotypic variation. Meanwhile, 2298 and 3509 DEGs were identified for JY93 and K6, respectively, after the SPB attack, and most of these genes were upregulated. Gene Ontology enrichment results indicated that the SPB-induced and differently expressed genes in both parents are involved in biological processes such as wound response, signal transduction, immune response, and phytohormone pathways. Interestingly, secondary metabolic processes such as flavonoid synthesis were only significantly enriched in the upregulated genes of JY93 after SPB chewing compared with K6. Finally, we identified 18 candidate genes related to soybean pod borer resistance through the integration of QTL mapping and RNA-Seq analysis. Seven of these genes had similar expression patterns to the mapping parents in four additional soybean germplasm after feeding by the SPB. These results provide additional knowledge of the early response and induced defense mechanisms against the SPB in soybean, which could help in breeding SPB-resistant soybean accessions.

## 1. Introduction

The soybean pod borer (*Leguminivora glycinivorella*) (SPB) is among the major injurious pests in northeast Asia and seriously affects soybean yield and quality [1,2]. Females oviposit on soybean plants in the summer (from mid-July to early August), then the larvae bore into the pods to feed on the seeds after hatching, reducing yields and affecting the quality of the seeds [3,4]. During field management of the soybean, controlling the SPB primarily relies on pesticide use. However, the long life cycle of the SPB and recurrent outbreaks require multiple applications of insecticides to control adult borers [3,5]. This approach not only increases farming costs but also causes pesticide residues in farmland and soybean products. In contrast, host plant resistance (HPR) is a better option for controlling the SPB. This is natural biotic stress resistance conferred by the genetic makeup of the plant, without the use of insecticides or transfer of foreign genes, to reduce insect losses [6,7]. Hence, it is essential to understand the molecular and genetic mechanisms of resistance in order to achieve favorable HPR.

Currently, most research has focused on utilizing transgenic methods to produce diapause factors in soybeans that can cause death or abnormal development of the SPB [3,5,8,9]. Fewer studies have been conducted to explore soybean’s native SPB resistance and responses to the SPB in vivo. Two previous studies on SPB resistance-related QTL mapping used recombinant inbred line (RIL) populations with SSR markers and identified three and four QTL loci [1,2]. Since the genetic background of the soybean germplasm is complex [10,11], those seven QTLs may not fully reveal the mechanism of SPB resistance in soybean. Research using different populations from previous reports could further explore the genetic mechanism of SPB resistance and contribute to obtaining stable QTLs for molecular marker-assisted selection to breed superior soybean cultivars. Meanwhile, although these studies provide unique insights into the mechanism of SPB resistance in soybean, the large intervals of QTL regions resulting from the low throughput of SSR markers make it difficult to obtain well-defined critical genes. High-throughput sequencing provides a large number of SNP markers to draw up a high-density linkage map covering the whole gene [12], and combined with reference genome information enables specific physical positioning of one marker on the chromosome, thereby facilitating access to all genetic information within a specific QTL region [13,14,15]. On the other hand, using transcriptomic data from tissues of population parents at specific times to obtain gene expression profiles can effectively resolve the excessive number of candidate genes in QTL intervals; for example, several studies used thousands or hundreds of genes of QTL intervals in which only a dozen or even a few candidate genes were retained after narrowing down the RNA-Seq for the mapping parents [16,17,18]. In addition, the differential transcription of genes between parents can be also used to decipher the molecular mechanisms of interesting traits [18,19].

The SPB feeds on developing soybean seeds inside pods [3,5]. Some studies have found that the thickness of the pericarp cell layer and the inner cell tissue layer in the soybean pod shell are negatively correlated with ease of entry into the pod by the SPB [20,21]. In addition, treating pods with methyl jasmonate and methyl salicylate increased the volatilization of alkanes and nitrogenous compounds. The number of adult SPBs was reduced in the area where the treated material was planted compared to the control, and the proportion of SPB larvae entering the pods was similarly reduced [22]. Therefore, the pod shell is the first layer and the key organ of defense against and response to SPBs in reducing their impact on soybean yield, but not seeds. However, previous studies have not explained how soybean resists SPB chewing in terms of the response of the pod shell, nor investigated the molecular basis of the response mechanism involved.

Plants have evolved complex and diverse biological mechanisms to resist insect damage [23]. Phytohormones, as signaling molecules, are responsible for the activation of transcription factors (TFs) and defense-related genes after triggering wound- or insect-specific elicitors [23,24,25]. In soybean infected with *Helicoverpa armigera*, the levels of jasmonic acid (JA) and salicylic acid (SA) were found to increase, which was lower in the insect-susceptible than in the insect-resistant material [26]. Inoculation and feeding of living soybean plants by stink bugs also induced the synthesis of JA, SA, and ethylene in pods and seeds [27]. At the same time, the WRKY family and wound-induced protein kinase genes of the resistant *Helicoverpa armigera* genotype were downregulated compared to the susceptible genotype [28]. As the first barrier against insect feeding, the cell wall, in addition to relying on the above-mentioned signals to transmit masticatory pressure, can also synthesize anti-insect-feeding substances, such as polygalacturonase-inhibiting proteins, to inhibit insect digestion of the cell wall, resulting in retarded insect growth [29]. Flavonoids are one of the main anti-insect compounds in soybean. Synthesis of genistein and rutin was induced in pods of insect-resistant varieties after feeding by *Piezodorus guildinii* [30]. Similarly, an infestation of *Nezara viridula* on soybean pods caused an increase in daidzein and genistein content in immature seeds, with significant differences in the levels of increase among five genotypes [31]. Zhao et al. [2] found that seed isoflavone content was associated with SPB resistance. Among the isoflavones daidzein, glycitein, and genistein, only the content of genistein was positively correlated with the number of seeds fed on by the SPB. The research also found that isoflavone-related QTLs overlapped with SPB-resistant QTLs. Further, Zhang et al. [32] found that the gene encoding UDP-glycosyltransferase (*Glyma.07g110300*) had a negative regulatory effect on isoflavonoid content in soybean and was associated with resistance to insects. The CRISPR/Cas9-mediated targeted mutagenesis lines showed significantly enhanced resistance to attack by *Helicoverpa armigera* and *Spodoptera litura*; however, the overexpression lines were sensitive to insect attack. Unfortunately, fewer genes or loci of insect resistance in soybean were discovered, and the molecular mechanism of SPB-related stress regulation has not been studied in depth.

As one of the major soybean-producing areas, northeastern China has been identified as a region with long-term infestation by the SPB, where there are abundant germplasm resources available. In the present study, SPB-resistant and SPB-sensitive accessions from cultivars in northeastern China, JY93 and K6, were selected as parents for manual crossing to establish the F_2_ population and a high-density linkage map was constructed using single nucleotide polymorphism (SNP) markers. A genome scan for SPB-resistant QTLs employing the genetic linkage map of the F_2_ population and two-year phenotypic data was performed. We also performed RNA-Seq analysis of the two parents to compare their gene expression changes after SPB feeding on pod shells to explore the early response mechanism of soybean to the SPB. Integrating QTL mapping and RNA-Seq data to identify key loci associated with SPB resistance could be beneficial for the genetic breeding of soybean with improved SPB resistance. These findings will add to our understanding of the underlying molecular basis of the mechanisms of SPB resistance in soybean.

## 2. Results

### 2.1. Phenotypic Variations in Eating Seed Percentage (ESP) for SPB

The phenotypic differences of crossing parents provided the basis for constructing the QTL population. *T*-tests showed significant differences in ESP between the two parents and the two years (*p*_2020_ = 0.0038, *p*_2021_ = 0.0009), with K6 being more prone to feeding by soybean pod borer than JY93, whereas for the same parent there was no significant difference in ESP between the two years (*p*_JY93_ = 0.3965, *p*_K6_ = 0.1364), which indicates the stability of parental traits (Figure 1A and Table 1). However, the phenotypic data of the F_2_ population differed significantly from the data of the F_2:3_ population (*t*-test, *p* = 0.0018), which may be due to the additional effect of genotypic segregation and the interaction of the new genotypes with the environment. In addition, the ESP of the two progeny populations was skewed toward JY93 in the mean and overall distribution and showed a trend of continuous distribution as well as a negative transgressive phenomenon (Figure 1 and Table 1). Moreover, the skewness and kurtosis of the offspring population’s phenotypic data indicated that the population conformed to the skew-normal distribution model in trait segregation and had the genetic characteristics of quantitative traits controlled by multiple genes (Figure 1B and Table 1). In conclusion, the phenotypic traits of the two populations were confirmed to meet the requirements for QTL mapping.

### 2.2. QTL Mapping Identified Four QTLs Controlling ESP

To obtain the genotypes of the F_2_ population, we sequenced all F_2_ individuals using genotyping by targeted sequencing (GTBS) [33] technology and obtained 33,578 SNPs (Appendix A). Then we filtered the SNPs by the following conditions: (1) different genotypes between two parents, (2) less than 20% missing rate, and (3) deleted redundancy marker by missing rate using BIN module of QTL IciMapping 4.0 [34]. Finally, a linkage map covering 4298.3 cM of linkage distance with an average distance of 3.03 cM between adjacent SNP markers was constructed using 1634 high-quality SNP markers (Table 2 and Appendix A). In addition, each chromosome was anchored with 24–155 markers with linkage distances of 136.37–348.46 cM. Although the distribution and average distance of SNP markers within the map vary across chromosomes, the average distance between markers is relatively small enough to satisfy the QTL mapping requirements.

With the combination of the F_2_ linkage map and the phenotypic data of the two populations, the QTLs underlying ESP were identified by the ICIM method. Four QTLs were identified with the logarithm of odds (LOD) ≥ 2.5 as the threshold [35], which were distributed on chromosomes 4, 9, 13, and 15 (Figure 2 and Table 3). Only *qESP-9-1* appeared in both years, but it is curious that its additive effects in the two years were in opposite directions (Table 3). Combined with the phenotypic differences in the population in the two years, we can speculate that ESP is strongly influenced by genotype and its interaction with the environment. Meanwhile, three QTLs were identified only in 2021. Among them, *qESP-4-1* had positive additive effects that were contributed by the maternal parent (JY93) alleles, and *qESP-13-1* and *qESP-15-1* showed negative additive effects that indicated phenotypic variations contributed by the paternal parent (K6) alleles (Table 3). Finally, we compared these QTLs with previous ESP-related results in the SoyBase database [36] and found that all four QTLs were newly identified. In addition, we relied on the physical positions of the SNP markers to target candidate genes within the four QTL intervals. Within the four QTL regions there were 75 (*qESP-4-1*), 2585 (*qESP-9-1*), 1023 (*qESP-13-1*), and 562 (*qESP-15-1*) candidate genes. Subsequently, the online WEGO 2.0 toolbox [37] was used to count GO annotation terms within all QTL loci based on the annotations of candidate genes (Appendix A). The results showed that cellular process ((GO:0009987), 2121 genes), metabolic process ((GO:0008152), 2031 genes), and response to stimulus ((GO:0050896), 1276 genes) were the three main GO entries for the 4320 genes in the four QTLs, and signaling ((GO:0023052), 377 genes) and immune system process ((GO:0002376), 221 genes) were also present in the four QTLs (Appendix A). Therefore, we speculate that the mechanism of soybean response to the SPB involves the above pathways.

### 2.3. Common Response Genes to SPB Feeding between Two Parents Involving Signaling, Defense, and Immune Biological Processes

Although GO annotation results for candidate genes within the QTL region suggested that stimulus response, signaling, and metabolic and immune processes are involved in soybean response to the SPB, we still have no insight into what happens within a pod when a SPB feeds on soybeans. Thus, for RNA-Seq, we collected a total of 12 pod materials from parents in the early eating stage according to SPB feeding trials (Figure 3A); the samples consisted of four experimental groups (three biological replicates within each group) divided into JY93 (SPB-resistant parent, not fed on by the SPB), JY93E (eaten by the SPB), K6 (SPB-susceptible parent, not fed on by the SPB), and K6E (eaten by the SPB). Then, RNA-Seq data were validated for gene expression by qRT-PCR (Appendix A), and 10 randomly selected genes from the QTL region were subjected to qRT-PCR, which showed a significant positive correlation with the results of RNA-Seq (*p* = 3.6 × 10^−6^, Pearson’s coefficient).

To identify response genes to SPB eating, gene expression was compared in pod shells before and after feeding treatment in the parental lines (JY93 and K6). The results show that many differently expressed genes (DEGs; (|log2 (fold change)| ≥ 1 and *p* adjust < 0.05)) were upregulated after SPB chewing, and the number was about two times higher than the number of downregulated DEGs in two parents after SPB chewing (Figure 3B). Meanwhile, 927 genes responded to SPB wounding in both parents (Figure 3B), and we termed them common response genes (CRGs). GO enrichment analysis of the CRGs revealed that 95 entries related to biological processes were significantly enriched (Benjamini–Hochberg method, *p* adjust < 0.05; Appendix A), and they were predominantly focused on processes related to regulating the response to stimulus (Figure 3C), including signaling, defense mechanisms, wounding response, and biological processes, and response mechanisms connected with immune or defense responses. Among them, the synthesis, metabolism, and response-related pathways of several defense hormones (JA, SA, and abscisic acid (ABA)) were significantly enriched (Appendix A), suggesting that three hormones play important roles in the SPB response. In addition, many metabolic pathways were significantly enriched by CRGs, mainly cell wall components (e.g., chitin and polysaccharides), sulfide metabolism (e.g., glycosinolate), and polyketide-related pathways (Appendix A), indicating that SPB mastication altered the metabolic processes in pod shells, and these substances may be involved in the mechanism of SPB resistance.

To further illuminate insect feeding-induced gene expression changes, we looked into the annotation information of CRGs (Appendix A). Similar to the GO enrichment results, genes related to signaling, reactive oxygen species (ROS), cell wall-associated processes, and JASMONATE-ZIM DOMAIN (JAZ) protein were identified. We also retrieved 108 transcription factors (TFs) after comparing the CRGs with data from PlantTFDB [38], among which the WRKY family (25 genes) was the most numerous, followed by the ERF (15 genes), bHLH (14 genes), MYB (12 genes), and NAC (12 genes) families. Interestingly, 25 disease-resistant proteins were also discovered. 

The expression trends of the genes in the four trial groups showed the following results: (i) In general, the genes of TFs represented by the WRKK family, signaling pathways, ROS-related enzymes, cell wall components, JA and ethylene synthesis, and secondary metabolite synthesis were upregulated after insect chewing. However, some genes showed differential expression in response to the SPB between the two parents, either induced expression in only one parent or significant differences in induced expression (Figure 3D and Appendix A). (ii) Some genes in the ERF, LBD, and HD-ZIP TF families and expansion-related metabolism were repressed (Figure 3D and Appendix A). (iii) Of the disease-resistant proteins, K6 was more sensitive to induction than JY93, i.e., the expression level of these genes was higher in K6E than in JY93E (Figure 3D). (iv) Although both JAZ protein and ROS-related genes were upregulated in both parents, the overall expression of JAZ protein was more upregulated in K6 than in JY93, and the reverse was the case for ROS-related genes (Appendix A). Altogether, we hypothesize that the early response of soybean to the SPB is as follows: the herbivorous behavior and certain secreted elicitors of the SPB activate cell wall or intracellular signaling receptors, then signaling pathways, to initiate biotic stress response and defense mechanism, whereas TFs have an important role in the response mechanism. Differential expression of genes involved in SPB resistance mechanisms between parents may underlie the differences in SPB resistance.

### 2.4. Flavonoid and Isoflavonoid Anabolism Are Responsible for the Variation in Parental Resistance to SPB

Differential response genes (DRGs) to SPB feeding between mapping parents may be involved in the resistance mechanism and are associated with differences in resistance. Here, we define DRGs as those DEGs occurring between before and after insectivory in only one parent, and 1871 and 2582 DRGs were obtained in JY93 and K6, respectively (Figure 3B). Of these, SPB feeding induced an upregulation of 1212 and 1870 genes and a downregulation of 659 and 712 genes, respectively, in JY93 and K6 (Appendix A). DRGs from both parents were classified into two categories following up- and downregulation and GO enrichment analysis. In combination with DRG annotation (Appendix A), as with CRGs, we identified biological processes such as signaling processes and defense responses associated with biotic stresses (Appendix A). The upregulated genes in the SPB-resistant parent JY93 were significantly enriched for several secondary metabolite processes and ABA response, whereas these GO terms were not enriched in the SPB-sensitive parent K6 (Figure 4A). Moreover, SA and xylan anabolism, sulfide transport, and chitin response were only enriched by upregulated genes in K6. Interestingly, whereas the downregulated genes in JY93 were not enriched in GO terms, processes associated with cell cycle and auxin were significantly enriched in K6 (Figure 4A).

Interestingly, flavonoid and its precursor biosynthetic pathways were significantly enriched only in the upregulated genes of the insect-resistant variety JY93 (Figure 4A); previous studies have shown a significant correlation between soybean seed isoflavone content and SPB resistance [2]. Hence, we conjecture that the secondary metabolite content in the pod shells is also involved in the SPB resistance mechanism. We selected relevant genes based on the annotation of DRGs to draw a heatmap following the expression levels of these genes (Figure 4B–E). It should be noted that the genes involved in polyketide biosynthesis are all chalcone synthases, which are key enzymes of the flavonoid biosynthesis pathway [39], and these genes overlapped completely with flavonoid biosynthesis in our results. The subsequent analysis focused on the flavonoid biosynthetic pathway because the polyketide pathway provides the upstream substrate for it [39]. Most of these genes were upregulated after the occurrence of SPB feeding, with the majority being highly expressed in JY93 compared to K6. Furthermore, the DEGs between K6E and JY93E were concentrated in genes related to flavonoid and isoflavonoid biosynthesis (Appendix A). Therefore, flavonoid and isoflavonoid anabolism may be important in the differential resistance to the SPB between JY93 and K6.

We also analyzed other responses to SPB-related biological pathways. Among these, several are similar to CRGs, such as response to chitin, i.e., most genes are induced to be upregulated by insect feeding and some genes differ significantly in expression between resistant and susceptible parents (Appendix A). Notably, the expression of genes associated with auxin synthesis was downregulated and that of catabolism-related genes was upregulated for DRGs between two parents, whereas the expression patterns of genes regulating ethylene synthesis and catabolism were the reverse (Appendix A). In addition, a large number of genes of the SAUR-like protein family were present in the biological process of response to auxin (Appendix A). These genes had completely opposite expression patterns in the two parents, with K6 being dominated by downregulated gene expression following insect attack and JY93 showing upregulation of most genes (Appendix A). For SA-related pathway genes, we found all genes to be calmodulin-related binding proteins or protein kinases (Appendix A). Moreover, the expression of these genes was sharply increased only in K6 after the induction of insect masticating. In summary, phytohormones are similarly involved in the mechanism by which soybean responds to SPB feeding, while possibly also defending against SPB feeding in a coordinated growth and defense approach. However, only a few of the genes of the above-mentioned biological processes were significantly differentially expressed between mapping parents in response to insect feeding (Appendix A). How they participate in SPB response and resistance and what role they play in resistance differences require further investigation.

### 2.5. Candidate Gene Prediction of ESP-Related QTLs Using Four Soybean Accessions

To reduce the candidate genes within the QTLs, intersection analysis was conducted with CRGs, DRGs, and all genes within the QTL regions. The four QTLs shared 81 and 364 genes with CRGs and DRGs, respectively (Appendix A). Among them, only 45 genes were differentially expressed in the two parents in response to SPB chewing, i.e., these are DEGs between K6E and JY93E. Meanwhile, among the 45 genes, 37 genes were mainly distributed in DRGs, with 8 candidate genes within CRGs. We then filtered 18 candidate genes with high expression from the above 45 genes by expression level (FPKM ≥ 5) [40] and gene annotation (Table 4). Among these genes, 13 candidate genes showed upregulation by SPB feeding, 7 genes showed higher expression in SPB-sensitive K6 compared to SPB-resistant JY93 genotype, 3 genes showed repressed expression (2 genes were repressed by SPB mastication only in JY93), and 2 genes were regulated in opposite patterns between the two parents (Table 4 and Appendix A). 

Furthermore, to verify whether the expression patterns of these candidate genes could be recurrent in other soybean germplasm, four varieties (two insect-susceptible and two insect-resistant) were selected for insect resistance assays under the same conditions as the mapping parents. The results showed that the expression patterns of only *Glyma.09G071600* and *Glyma.09G072000* were consistent in the mapping parents and the four germplasm (Figure 5). Meanwhile, among the resistant varieties, *Glyma.09G128400* and *Glyma.09G241800* showed similar expression patterns to the resistant parent JY93; among the insect-susceptible varieties, *Glyma.09G131900*, *Glyma.09G204500* and *Glyma.13G053600* showed similar expression trends to the insect-susceptible parent K6 (Figure 5). Therefore, we suggest that the above seven genes play a stable and important role in soybean resistance to the SPB.

## 3. Discussion

### 3.1. QTL Mapping Based on SNP Markers Combined with RNA-Seq for Efficient and Rapid Identification of QTLs and Candidate Genes Related to SPB Resistance

The soybean pod borer is known to be among the major pests in the soybean-producing regions of northeast Asia [1,2], where numerous germplasms have been used for breeding superior varieties because they are free of or less susceptible to SPB injury. It is therefore necessary to explore the genetic mechanisms of SPB resistance based on the major cultivars of northeast China. Although sequencing-based SNP markers have been widely used for QTL mapping of some important phenotypes in soybean [13,15,41], they have been rarely used to search for QTLs related to SPB resistance. In this work, we used several thousand polymorphic SNP markers to construct a linkage map and perform QTL mapping in the F_2_ population. As a result, a linkage map covering 20 chromosomes with a total length of 4298.30 cM and an average marker distance of 2.63 cM was obtained (Table 2 and Figure 2). Four QTLs associated with SPB resistance were identified that had not been found in previous studies (Table 3). Among them, *qESP-9-1* was the stable QTL identified in all environments. However, both previous studies about SPB resistance were based on SSR markers for QTL mapping in RIL populations. The average distance between the obtained linkage map markers was about 10 times higher than in this study, and the breeding of the RIL population took 5 or 6 years [1,2]. As a control, our study obtained F_2_ and F_2:3_ populations and scanned the stable QTL in only 3 years, whereas transcriptome data from pod shells were used to screen 4245 genes in four QTL intervals to 18 candidate genes, significantly narrowing down the range of genes requiring further validation. Therefore, by using the combination of QTL mapping under high-throughput sequencing and RNA-Seq analysis of population parents at specific periods, we can quickly and effectively obtain QTL and candidate genes for traits of interest [16,17,18]. Furthermore, using qRT-PCR, we found that only 7 of these 18 genes could show similar expression patterns in other soybean accessions. In particular, *Glyma.09G71600* and *Glyma.09G07200*, their expression changes were completely consistent with the two mapping parents. Therefore, these genes should be steadily involved in the mechanism of soybean resistance to SPB. However, the specific functions of these genes need to be further investigated.

In addition, it is noteworthy that the QTL *qESP-9-1* did not have the same direction of an additive effect in 2 years. Although the mean values of the F_2:3_ lines were used for QTL mapping, the genotypes of the F_2:3_ population were segregated, resulting in large changes in genotypes and their interaction effects with the environment; previous studies also showed differences in the distribution of phenotypic data for ESP in the F_2_ and F_2:3_ populations [4]. The variation in parental ESP data at 2 years also suggests that SPB feeding on soybean in the natural environment may have been influenced by the environment (Figure 1 and Table 1). In addition, for the same genotype, environmental changes can sometimes change the direction of QTL additive effects [42]. In summary, *qESP-9-1* had a specific additive effect in two environments but is still a stable QTL that will be useful for marker-assisted breeding for SPB resistance in soybean.

### 3.2. SPB Chewing-Responsive DEGs in Soybean Pod Shells Associated with Signal Transduction, Function Proteins, Secondary Metabolite Pathways, and TFs

Transcriptome analysis can indicate global expression profile alterations in plants under biotic stresses, thus helping to identify differentially expressed genes with their transcript abundances and regulatory mechanisms [43,44]. In this study, 2298 and 3509 DEGs were identified for JY93 and K6, respectively, after an attack by an SPB, and most of these genes were induced to be upregulated by the SPB (Figure 3B and Appendix A). Some of these genes are involved in Ca^2+^ signaling and MAPK cascade pathways, which supports previous studies that found the two pathways involved in insect resistance mechanisms in soybean [45,46,47]. Meanwhile, the initiation of Ca^2+^ signaling may be associated with glutamate mobilization after insect chewing [48]. The candidate gene *Glyma.09G131900* (glutaminyl cyclase) may play a role in the soybean SPB resistance mechanism by depleting glutamate [49]. Curiously, many genes related to signaling molecules in the leucine-rich repeat (LRR) protein family, which is commonly found in plants responding to pathogen invasion [50,51], are also activated by SPB chewing (Appendix A). The expression patterns of several disease-resistant proteins are also similar to those of LRR genes (Figure 3D and Appendix A). 

Considering that plants are anchored to the land and thus have had to develop a complex set of mechanisms to resist attack by different pathogens and insects at certain periods [52,53,54], we believe that the resistance of soybean to the SPB also interacts with other biological resistance [55]. The candidate gene *Glyma.13G053600* that can be stably activated by the SPB in insect-susceptible accessions is a member of the Malectin family, which has been shown to be associated with plant resistance to fungi [50] (Figure 5). Furthermore, only two pathogenesis-related proteins of the 18 genes in the initial selected were induced to be upregulated by the SPB in JY93, whereas the rest of these genes were upregulated in K6 after being subjected to SPB feeding (Appendix A). These results suggest that the operation of most non-SPB resistance mechanisms may reduce the resistance of soybean to the SPB [56,57]. Moreover, our search of the public soybean transcriptome database [58] revealed that the 18 candidate genes had high expression in response to multiple biotic stresses (Appendix A), which supports the existence of a reciprocal network of resistance to biotic invasion in soybean. However, further validation is needed to understand the specific mechanism by which soybean resists the invasion of multiple organisms at the same time.

The enrichment analysis of DEGs showed that in addition to signaling, secondary metabolites play an important role in soybean resistance to SPB feeding, especially cell wall component anabolism and phenylpropanoid biosynthesis (Appendix A). It has been shown that cell wall components can induce the production of toxic substances in plants to enhance their resistance to insects [59]. Phenylpropanoid is the substrate for the synthesis of many anti-insect substances, which can eventually be changed into two major substance groups [60]: lignin, related to cell wall components, and flavonoids, which will be discussed later in this section. Meanwhile, earlier studies showed that TFs can regulate plant defense against insect herbivore response through interactions with plant hormones and mediate the metabolism of secondary substances [28,61]. According to the results of TF identification, 561 TFs, including WRKY, MYB, NAC, bHLH, ERF, and other families, were up- or downregulated by both parents after SPB feeding (Appendix A). Some of these genes were selected as candidates because of their association with defense hormones and flavonoid synthesis (Table 3) and will be discussed below.

### 3.3. Phytohormone Cross-Talk and Flavonoid Metabolism May Mediate Soybean Response to SPB and Resistance Mechanisms

Plant hormones are activated by insect feeding pressure and actively regulate many inducible plant defense systems [25,53]. In our work, the terms enriched in CRGs and DRGs involved many plant hormones, including JA, SA, ABA, ethylene, and auxin (Figure 3C,D and Appendix A). Interestingly, the transcription of synthesis-related genes of JA and ethylene, which are known to improve soybean’s ability to resist insects [62,63,64,65,66], was significantly enhanced (Appendix A). The catabolic process genes of auxin are upregulated and its synthesis-related genes are repressed (Appendix A). We also identified two candidate genes whose TFs are regulated by hormones. One is the HD-ZIP family gene *Glyma.09G241800*, which was stable and downregulated only in the insect-resistant accessions after an SPB attack (Figure 5), whereas its sunflower homolog could be an inhibited expression by SA [67]. The expression pattern of *Glyma.09G072000*, another ERF family gene, is regulated by ethylene, which was distinct between the SPB-resistant varieties with differences after SPB feeding (Figure 5), and its homolog in *Arabidopsis* interacted with *IAA5* to regulate stress response [68]. The above results suggest that TFs are involved in phytohormone cross-talk and related to the SPB resistance mechanism of soybean [28,69].

The expression patterns of the core module genes of JA signaling [70] were significantly different in the two parents. Among them, *COI1* (*Glyma.14G062100*) showed unaltered expression in SPB-resistant cultivar JY93, whereas its expression in SPB-susceptible cultivar K6 was repressed after SPB feeding (Appendix A). The expression pattern of *MYC2* (*Glyma.09G204500*), one of the candidate genes, was induced to be upregulated by insect feeding in all susceptible SPB germplasm (Figure 5). All 17 JAZs containing *Glyma.09G071600* were activated by the SPB in all soybean accessions; however, all genes were more highly expressed in K6 than JY93 (Appendix A). As the binding of JAZ to MYC2 inhibits the activation of defense-related genes regulated by JA [71,72,73,74], we speculated that the response mechanism of K6 after SPB chewing is repressed by exciters in the SPB, whereas *MYC2* enhances transcription to induce high levels of JAZ in the nucleus for the activation of defense-related genes [74]. At the same time, JY93 should respond to the SPB faster than K6 based on the fact that the expression of *MYC2* in JY93 does not differ significantly before and after SPB feeding. Although these conclusions need to be confirmed through genetic transmission and further studies, the transcriptional differences in these genes between the two parents suggest that JA-related pathways play an important role in soybean resistance to the SPB. 

On the other hand, the high expression of synthesis genes of JA and ethylene implies that the hormone levels increase in plants. Previous studies have shown that JA and ethylene can also increase the production of flavonoids [27,75]. These substances have been shown to enhance resistance against several herbivore insects in soybean [26,27,30,31], and Zhao et al. [2] revealed a significant correlation between flavonoid content in soybean seeds and the seed-eating rate of the SPB. We observed that flavonoid-related genes were activated in soybean pod shells after SPB mastication (Appendix A), and most of these genes were expressed at higher levels in JY93 than in K6 after SPB feeding (Figure 4B,D). Among them, *Glyma.09G128400* was identified as a candidate gene because it has been shown to activate the synthesis of epicatechin, one of the flavonoids [76]; this gene had a high transcript level in JY93 only after SPB chewing (Figure 5A). Unfortunately, among the four germplasm additionally utilized, the gene expression level of the sensitive variety S2 was much higher than the other materials after being fed by an SPB. Considering that qRT-PCR is not an absolute quantification of gene expression and the complexity of the resistance mechanism, this gene is solidly involved in the SPB resistance process, but the specific mechanism needs to be further explored. Moreover, the UDP-glucosyltransferase family, to which *Glyma.09G128400* belongs, was also revealed to impact the ability of soybeans to resist insects [32]. Thus, the above results suggest that the flavonoid content of soybean pod hulls affects the ability of soybean to resist the SPB. However, further studies are needed to know exactly which flavonoid substances play a major role and how they resist the SPB.

## 4. Materials and Methods

### 4.1. Soybean Plant Materials and Growing Environment

A total of 232 F_2_ mapping populations were developed from a cross between Jiyu 93 (JY93, SPB-resistant) and Kangxianchong 6 (K6, SPB sensitive) at the experimental field of Jilin Agricultural University (Changchun, China). JY93 is one of the main cultivars in northeast China, whereas K6 is a local cultivar in Heilongjiang Province. To improve the power of QTL mapping [77], the phenotypic data of the F_2:3_ population obtained by the single seed descent method using the F_2_ lines were also used for QTL mapping, considering the low heritability of SPB resistance in soybean [4]. All crossing progeny and their parents were planted at the experimental field of Jilin Agricultural University. The type of soil at the field is loam, and standard field management practices were followed. In April 2020 and 2021, materials were sown in fields in single-row plots 65 cm wide with 4 m row length and 10 cm spacing between plants. It should be noted that the two parents and the F_2:3_ population were planted in one line per row, the F_2_ population was sown as a single plant per seed, and the parents were planted in two rows per year, F_2_ in 2020 and F_2:3_ in 2021.

In 2021, the two parents of the mapping population were subjected to insect feeding trials under the above-described conditions, which were divided into an insect feeding group and a control group. Both groups were covered with insect-proof nets at the same time in early July, and 9 soybean pod borers were placed inside the net of the insect-receiving group in early August. The insect-eaten pods were picked out 10 days after placing the SPB, whereas healthy pods were randomly selected from the control group without an SPB infestation. The selected pods were immediately stored in liquid nitrogen at −80 °C for subsequent RNA sequencing. Each group of 2 parents had 3 biological replicates, with 2–3 insect-eaten or control pod shells taken from each replicate. The 4 groups of 12 samples were recorded as JY93 (control, not fed on by SPB), JY93E (chewed by SPB), K6 (control, not fed on by SPB), and K6E (chewed by SPB) according to the parents and treatments. In addition, two SPB-resistant germplasm [ZDD00651(R1) and ZDD08352(R2)] and sensitive germplasm [ZDD14240(S1) and ZDD00023(S2)] were grown in the net chamber, these pod shells from the four varieties are also collected and used for validation of candidate genes. The code of these samples is similar to that of the two parents, e.g., R1 (control, not fed on by SPB) and R1E (chewed by SPB).

### 4.2. Determination of Eating Seed Percentage for SPB 

After the soybean plants matured, single plant materials were collected from the F_2_ population, and 5 consecutive and uninterrupted plants were collected from F_2:3_ in rows corresponding to each line, with the same parents as the F_2:3_ population but with two rows collected from each parent. The number of seeds per plant and insect-feeding seeds per plant was recorded for the collected materials indoors. The ESP in the natural growing environment was calculated according to the following formula: P = n/N × 100%, where P is ESP, n is number of insect-eaten seeds per plant, and N is number of seeds per plant. The ESP for each line of F_2:3_ is the average of 5 plants from the corresponding row. Descriptive statistics for ESP among the mapping population and the parental lines were calculated using DPS v9.5 software [78]. Considering the sample size and the effects of the field environment on ESP, Student’s *t*-test was used to test the differences between any two group samples [79]. In addition, *t*-test was performed using the t.test command in R (r-project.org). The R package ggpubr was used to visualize the phenotypic data.

### 4.3. SNP Identification by Sequencing

Genomic DNA from fresh leaves of the F_2_ mapping population and their parents was isolated using the Genomic DNA Extraction Kit (DP305-03; Tiangen Biotech Co., Ltd., Beijing, China). The GTBS method [33] was used to construct genome sequencing libraries (Mol Breeding Biotech Co., Ltd. Shijiazhuang, China). Briefly, high-quality plant genomic DNA is digested by methylation-sensitive endonucleases after the quality and concentration of DNA is examined using agarose gel electrophoresis and NanoDrop ND-1000 (NanoDrop, Wilmington, DE, USA). The target DNA fragments are later captured by designed biotin markers and finally retrieved for PCR amplification using the specific primers to obtain the sequencing library. The sequencing data from the MGI2000 platform were quality-controlled by FastQC v0.20.0 (www.bioinformatics.babraham.ac.uk/projects/fastqc, accessed on 6 May 2010) and then aligned to the Wm82.a2.v1 reference genome from phytozome [80,81] using BWA v0.7.17 (bio-bwa.sourceforge.net/, accessed on 7 November 2017). GATK v3.5 (software.broadinstitute.org/gatk, accessed on 7 June 2019) was used for SNP identification, and finally, the genotyping files were formed by extracting all SNP information using custom Perl scripts.

### 4.4. Construction of Linkage Map and QTL Localization

First, SNP markers with identical genotypes at the parental loci were filtered and the SNP information was transcoded as A (identical to the maternal loci), B (identical to the paternal loci), H (heterozygous genotypes), or X (genotypic missing). After that, linkage analysis was conducted using QTL IciMapping 4.0 [34]. SNP marker with ≥20% deletion rate was removed under the BIN function of the software, and redundant markers were screened out according to the deletion rate. Finally, the MAP function was applied to construct the linkage map based on the nnTwoOpt algorithm, with reference to Zhang et al. [82]. The R package r/qtl [83] was used to visualize the linkage map. The genetic mapping of F_2_ and the phenotypic value of crossing offspring were combined to perform QTL analysis with QTL IciMapping 4.0. Inclusive composite interval mapping under an LOD threshold of 2.5 was carried out with default settings to compute QTL information [35], including LOD score, percent phenotypic variance effects (PVE), and additive effects.

### 4.5. RNA Isolation and RNA-Seq Data Analysis

Total RNA from 12 samples of pod shells according to the insect feeding trials were isolated and purified using TRIzol (15596018, Thermo Fisher, Waltham, MA, USA) following the manufacturer’s procedure. The quantity and purity of total RNA were analyzed by a NanoDrop ND-1000 (Thermo Fisher, Waltham, MA, USA) and a Bioanalyzer 2100 (Agilent, Santa Clara, CA, USA). Finally, we performed 2 × 150 bp paired-end sequencing (PE150) on an Illumina NovaSeq 6000 following the vendor’s recommended protocol for all samples (LC Bio Technology Co., Ltd., Hangzhou, China). Using Cutadapt [84] to gain clean data, for each sample, ≥6 Gb of data was produced, for which the quality-control metric Q30 was >96%. Valid data from all samples were aligned by HISAT2 [85] to the reference genome used for DNA-Seq, the resulting data were used to calculate mRNA expression levels using StringTie [86], and the expression levels were normalized to fragments-per-kilobase of transcript-per-million mapped reads (FPKM).

### 4.6. Bioinformatic Analysis

Annotation information for all soybean genes was obtained from the SoyBase [36] and Phytozome [80] databases. According to the instructions, DESeq2 [87] was used to analyze differentially expressed genes (DEGs) between trial groups (|log2 fold change| ≥ 1 and *p* adjust < 0.05). clusterProfiler 4.0 [88] (Benjamini–Hochberg method, *p* adjust < 0.05) was used to find significantly enriched GO terms in the input list of DEGs.

### 4.7. Quantitative Reverse Transcription PCR (qRT-PCR)

qRT-PCR was used for assessing the quality of the transcriptome and verifying the expression of candidate genes in other materials. First, the RNA samples for RNA-Seq were reverse transcribed to cDNA using Maxima Reverse Transcriptase (EP0743, Thermo Scientific, Waltham, MA, USA). Then the PCR mix was prepared using 2× SG Fast qPCR Master Mix reagent (B639271, Roche, Rotkreuz, Switzerland) according to the manufacturer’s instructions. Finally, a LightCycler 480 II PCR machine (Roche, Rotkreuz, Switzerland) was used for qRT-PCR amplification with gene-specific primers, listed in Appendix A. The actin gene (*Glyma.18G290800*) was used as the housekeeping gene and the relative expression of each gene was calculated by the 2^−∆∆Ct^ method [89]. Pearson’s correlation coefficient was used to measure the degree of correlation between RNA-Seq and qRT-PCR results; the coefficient, significance of the coefficient, and visualization were conducted using the online bioinformatics toolbox (https://www.bioinformatics.com.cn, accessed on 1 June 2022). Correlations were considered significant if the *p*-value of the correlation coefficient was less than 0.05.

## 5. Conclusions

In the present study, we identified four ESP-related QTLs on chromosomes 4, 9, 13, and 15, among which qESP-9-1 was the stable QTL, which was detected in two years. Expression profiles of damaged pod shells showed significantly altered expression of numerous genes in soybean after SPB feeding; 2298 and 3509 DEGs were identified for JY93 and K6, respectively, most showing an upregulation trend. Meanwhile, biparentally shared response genes were mainly enriched in stimulus-responsive biological processes such as wound response, signal transduction, and immune response. Among them, the synthesis of several defense hormones (JA, SA, and ABA) was significantly enriched, which may be a necessary pathway in response to the SPB. Interestingly, the secondary metabolic pathways, mainly flavonoid metabolism, were significantly enhanced only in the insect-resistant parents among the differentially responsive genes of both parents. These results suggest that hormonal and secondary metabolic processes play an important role in soybean resistance to the SPB. Combining the results of QTL mapping and RNA-Seq, the 4245 genes within the four QTL regions were narrowed down to 18 genes. Seven of these genes were able to show a similar expression pattern in other soybean germplasm as in the mapping parents. In short, these results not only enhance our understanding of soybean response to the SPB and defense mechanisms but also provide a reference for marker-assisted selection in conjunction with phenotyping of breeding populations or individuals to accelerate the selection of superior varieties to tackle yield losses from the SPB in northeastern China.

## Figures and Tables

**Figure 1 ijms-23-10910-f001:**
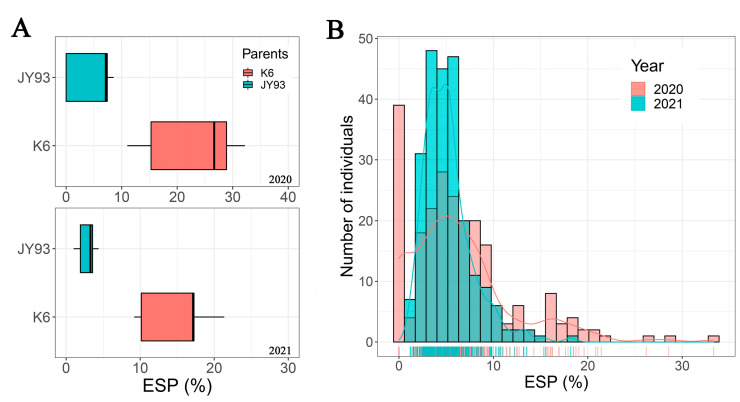
(**A**) Distribution of ESP between two parents: phenotypic data in 2020 (top graph) and 2021 (bottom graph). (**B**) Frequency distribution of ESP and statistical analysis of offspring populations: X-axis is ESP (%) and vertical line on the axis is distribution of population data, and Y-axis is count of population distribution.

**Figure 2 ijms-23-10910-f002:**
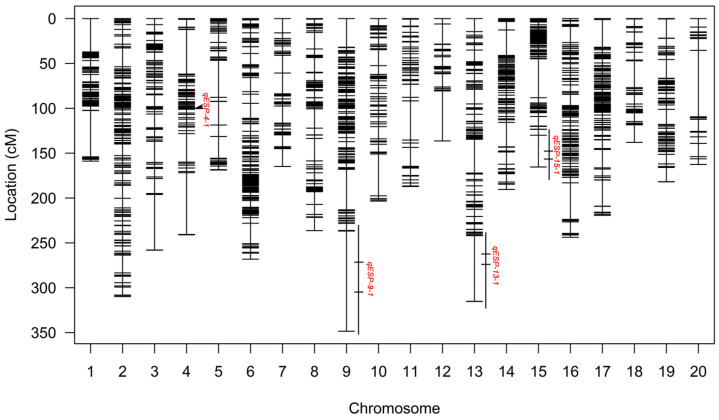
Genetic map of F_2_ population and ESP-related QTLs. Ruler on left side indicates interval distance between SNP markers using centimorgan (cM) as the unit, and ESP-related QTLs are annotated to the right of chromosomes. Line at the right of some chromosomes indicates physical QTL region, and horizontal line inside one line indicates confidence interval of QTL region.

**Figure 3 ijms-23-10910-f003:**
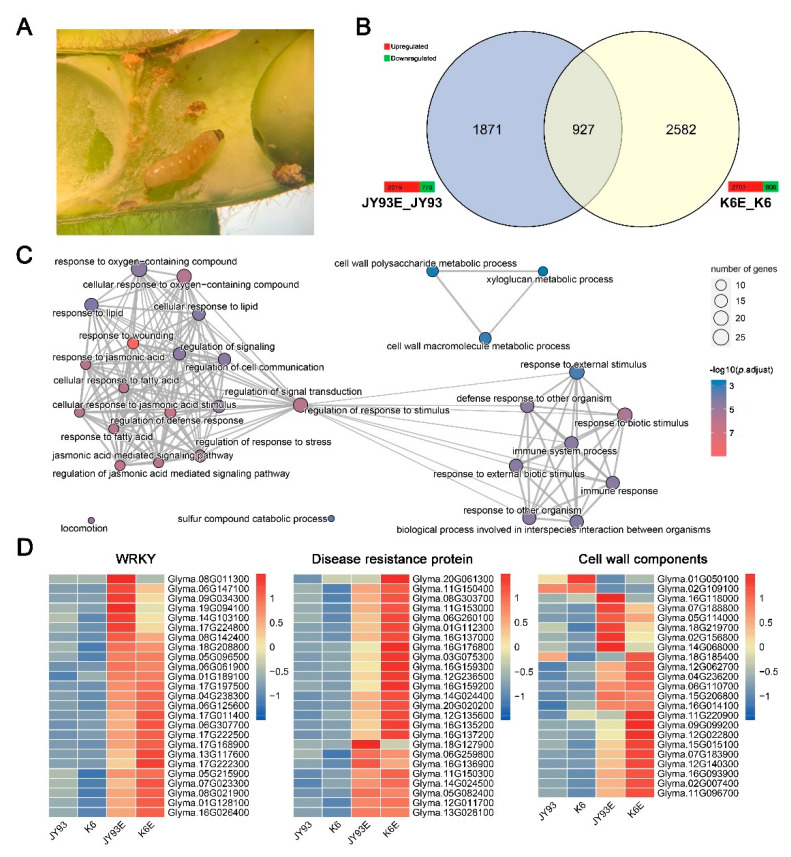
CRGs of two mapping parents. (**A**) State of SPB at time of pod collection (magnified 25× under dissecting microscope). (**B**) Statistical analysis of DEGs of two parents before and after SBP eating. Color bars at edge of Venn diagram indicate number of DEGs; red indicates upregulation and green indicates downregulation. (**C**) GO enrichment analysis of CRGs using clusterProfiler 4.0. Legend on right side indicates number of genes enriched (top) and *p* adjust value of enriched entry (bottom). Larger circles in graph indicate higher number of genes; redder colors indicate greater significance. (**D**) Expression levels of representative CRGs. Color bar on right side indicates expression levels of certain genes by Z-score method; red is high expression, blue is low expression.

**Figure 4 ijms-23-10910-f004:**
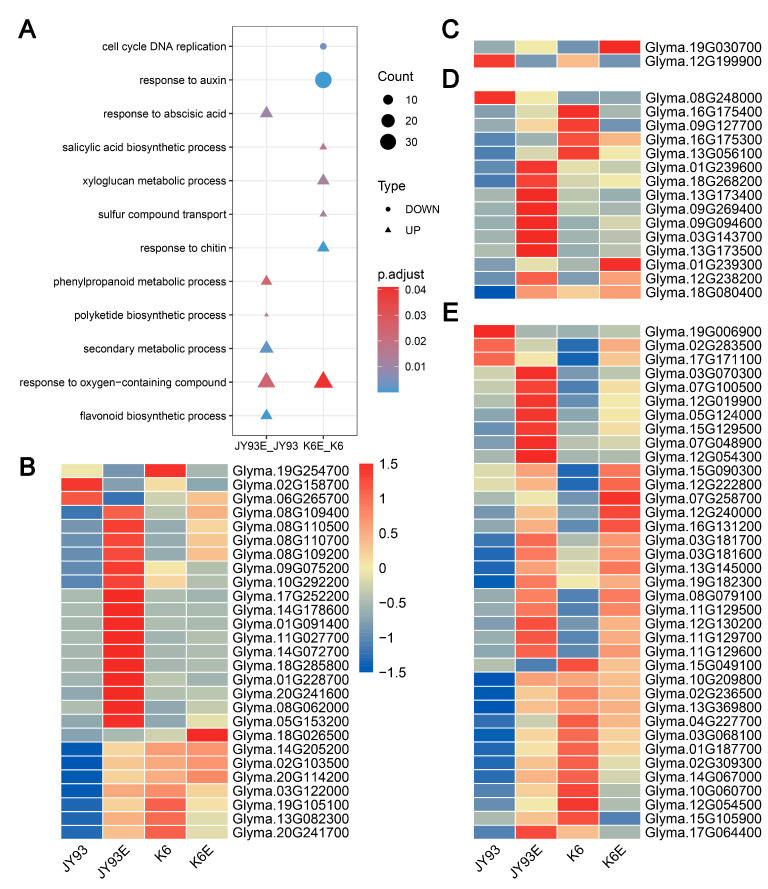
DRGs between mapping parents JY93 and K6. (**A**) GO enrichment analysis of DRGs. Three legends on right side show, from top to bottom, number of enriched genes (larger shape indicates greater number), type of gene response (circles indicate suppressed expression and triangles indicate upregulated expression after feeding by SPB), and significantly enriched level of entries (redder color indicates higher significance). (**B**–**E**) Expression levels of flavonoid metabolism-related genes: flavonoid metabolism, flavone and flavanol metabolism, isoflavonoid metabolism, and phenylpropanoid metabolism, respectively. Color bar on right side indicates expression levels of certain genes by Z-score method; red is high expression, blue is low expression.

**Figure 5 ijms-23-10910-f005:**
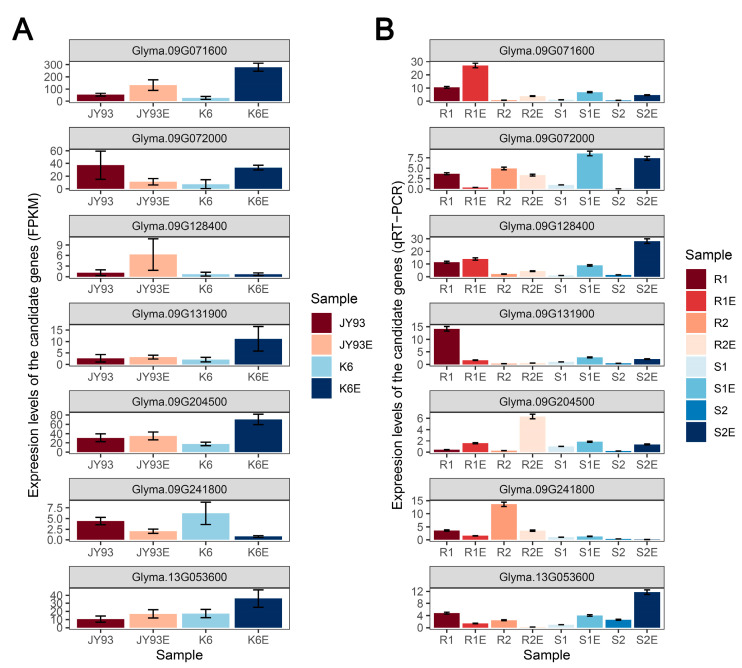
Expression levels of all candidate genes. (**A**) RNA-Seq data of the candidate genes in the mapping parents. The scale on the left indicates the expression levels (FPKM). (**B**) qRT-PCR data of the candidate genes in the four soybean varieties. Among these, two of which were resistant varieties [R1 (ESP = 2.48%, *n* = 15), R2(ESP = 2.49%, *n* = 15)] and the other two were susceptible varieties [S1(ESP = 16.00%, *n* = 15), S2 (ESP = 17.59%, *n* = 15)]. The scale on the left indicates the relative expression levels (2^−∆∆Ct^).

**Table 1 ijms-23-10910-t001:** Statistical analysis of hybrid population and its parental ESP.

Year	Parents		Offspring of JY93 × K6						
	JY93 ^+^	K6 ^+^	Min	Max	Mean	SD	CV (%)	Skewness	Kurtosis
2020	4.60 ^A,a^	22.81 ^B,a^	0.00	33.33	6.44 ^b^	5.63	87.39	1.48	3.17
2021	2.84 ^A,a^	15.02 ^B,a^	1.19	18.18	5.19 ^a^	2.65	51.09	1.48	3.38

^+^ Mean ESP of JY93 and K6 calculated from 10 plants per parent in two rows. SD, standard deviation; CV, coefficient of variation. Same letter does not indicate significant differences; upper case represents between parents in the same year, lower case represents between the same parents or different progeny groups between two years. All statistical analyses were followed by Student’s *t*-test using 99.5% confidence interval, thus *p*-value < 0.05 is considered to be significantly different.

**Table 2 ijms-23-10910-t002:** Total SNP numbers and linkage distances of chromosomes in F_2_ population.

Chromosome	Linkage Group	Number of SNPs	Linkage Distance (cM)	Average Distance between Markers (cM)
Chr01	D1a	69	158.87	2.30
Chr02	D1b	155	309.88	2.00
Chr03	N	78	258.15	3.31
Chr04	C1	68	241.04	3.54
Chr05	A1	45	168.80	3.75
Chr06	C2	147	268.14	1.82
Chr07	M	58	164.83	2.84
Chr08	A2	82	236.34	2.88
Chr09	K	135	348.46	2.58
Chr10	O	60	203.45	3.39
Chr11	B1	51	187.19	3.67
Chr12	H	28	136.37	4.87
Chr13	F	90	315.31	3.50
Chr14	B2	95	190.43	2.00
Chr15	E	87	165.41	1.90
Chr16	J	120	243.79	2.03
Chr17	D2	122	219.30	1.80
Chr18	G	43	137.98	3.21
Chr19	L	77	181.97	2.36
Chr20	I	24	162.59	6.77
Total	−	1634	4298.30	2.63

**Table 3 ijms-23-10910-t003:** Analysis of ESP-related QTLs in offspring population from JY93 × K6.

QTL Name	Year	Chr. ^a^	Position (cM)	LeftCI	RightCI	LOD	Interval Region (bp) ^c^	PVE	Additive Effect
				(cM) ^b^	(cM) ^b^			(%)	
*qESP-4-1*	2021	4	105	100.5	107.5	3.24	8,394,899–9,376,933	0.8439	0.0059
*qESP-9-1*	2020	9	284	271.5	293.5	8.05	2,021,437–49,893,816	4.3313	−0.0632
*qESP-9-1*	2021	9	292	276.5	309.5	2.64	2,021,437–49,893,816	6.0935	0.0231
*qESP-13-1*	2021	13	273	268.5	275.5	3.16	14,613,981–26,717,362	5.7935	−0.0219
*qESP-15-1*	2021	15	165	149.5	165	2.82	583,964–4,836,857	0.7931	−0.0013

^a^ Chromosome. ^b^ Genetic border of 95% confidence interval for detected QTLs. ^c^ Physical interval region of 95% confidence interval for detected QTLs.

**Table 4 ijms-23-10910-t004:** Candidate genes of four ESP-related QTLs.

Gene Locus	QTL Name	*A. thaliana* Homolog	Response Pattern ^a^	Description
Glyma.04G098000	*qESP-4-1*	AT4G40080	Reverse	ENTH/ANTH/VHS superfamily protein
Glyma.09G040500	*qESP-9-1*	NA	Up (JY93)	Pathogenesis-related protein
Glyma.09G062900	*qESP-9-1*	AT4G26140	Up (JY93)	Beta-galactosidase
Glyma.09G071600	*qESP-9-1*	AT1G19180	Co-up	Jasmonate-zim-domain protein
Glyma.09G072000	*qESP-9-1*	AT1G19210	Reverse	ERF family proteins
Glyma.09G128400	*qESP-9-1*	AT3G16520	Up (JY93)	UDP-glucosyl transferase
Glyma.09G131900	*qESP-9-1*	AT4G25720	Up (K6)	Glutaminyl cyclase
Glyma.09G143800	*qESP-9-1*	AT1G63120	Up (K6)	RHOMBOID-like protein
Glyma.09G149400	*qESP-9-1*	AT5G62180	Down (JY93)	Carboxylesterase
Glyma.09G204500	*qESP-9-1*	AT1G32640	Up (K6)	MYC family protein
Glyma.09G241800	*qESP-9-1*	AT3G60390	Co-down	HD-ZIP family proteins
Glyma.13G053600	*qESP-13-1*	AT3G51550	Up (K6)	Malectin family protein
Glyma.13G054200	*qESP-13-1*	AT3G51550	Up (K6)	Malectin family protein
Glyma.13G054400	*qESP-13-1*	AT3G51550	Up (K6)	Malectin family protein
Glyma.13G054600	*qESP-13-1*	AT4G20050	Up (JY93)	Pectin lyase-like superfamily protein
Glyma.13G113800	*qESP-13-1*	AT5G20820	Up (JY93)	SAUR-like auxin-responsive protein family
Glyma.15G024800	*qESP-15-1*	AT5G17680	Down (JY93)	Disease-resistant protein
Glyma.15G062300	*qESP-15-1*	AT2G19990	Up (JY93)	Pathogenesis-related protein like

^a^ Reverse indicates candidate gene expression pattern after SPB feeding of two parents differs; Co-up and Co-down indicate simultaneous upregulation or downregulation of candidate genes of two parents after SPB feeding; Up/Down (JY93) and Up/Down (K6) indicate that the candidate gene in only one parent was upregulated or downregulated after SPB feeding.

## Data Availability

All RNA-Seq data can be found at the NGDC website (https://ngdc.cncb.ac.cn/), bioproject PRJCA010033.

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
