# Peer review of "Identifying Soybean Pod Borer (Leguminivora glycinivorella) Resistance QTLs and the Mechanism of Induced Defense Using Linkage Mapping and RNA-Seq Analysis"

_ijms, 2022, doi:10.3390/ijms231810910_

Round 1

Reviewer 1 Report

Very good work and nice presentation

Author Response

We are very grateful for your approval of this article, and we have invited native English-speaking colleague to proofread and polish the language in the revised manuscript according to your suggestion. Thank you again for your review!

Reviewer 2 Report

1. The authors use linkage mapping and RNA-Seq analysis to identify genes involved in the defense system of soybean against Leguminivora glycinivorella, and eventually, they found 18 candidate genes. After reading this manuscript, I think the major problem of this study is the experiments are still too preliminary and did not get anything about the in-depth mechanism. The authors need to further validate the roles of the 18 candidate genes and find key genes that play the chief role in the defense system.

2. M&M: The authors should describe more details for the experiential design, particularly the number of replicates for all the quantitative data.

3. There is a lack of a statistics section in M&M, and all the results should be described according to statistics.

4. L30: a large up-regulated genes?

5. L36: through integration of QTL mapping and RNA-Seq analysis - through the integration of QTL mapping and RNA-Seq analysis

6. L68: hundreds genes - hundreds of genes

7. L85: for activation - for the activation

8. L399: Many these genes - Many of these genes

9. L401: to transcript - to the transcript

10. L482: that as one of flavonoids - that is one of the flavonoids

11. L600: for each trail groups - for each trail group

12. The authors need to revise grammar errors throughout the manuscript.

Author Response

Thank you for your constructive comments and suggestions for improving this article. We have revised the manuscript in accordance with your valuable suggestions, which can be found in the revised manuscript.

Point 1: The authors use linkage mapping and RNA-Seq analysis to identify genes involved in the defense system of soybean against Leguminivora glycinivorella, and eventually, they found 18 candidate genes. After reading this manuscript, I think the major problem of this study is the experiments are still too preliminary and did not get anything about the in-depth mechanism. The authors need to further validate the roles of the 18 candidate genes and find key genes that play the chief role in the defense system.

Response 1: Thank you very much for reviewing this article and provided valuable suggestions. We agree with your point that candidate genes can only be used in breeding programs when they have been verified for defined functions. These 18 candidate genes are highly expressed in the pod shell after the SPB larvae chewing, and most of them are derived from the stable SPB-related QTL qESP-9-1. In the subsequent research, we would use transgenic and gene editing techniques to further investigate the specific functions of these genes. Since most of the genes may be associated with hormones and metabolites, we also intend to perform metabolomic analysis of soybean pod hulls and seeds and combine this with the relative expression of 18 candidate genes at the time of sampling to further reveal the mechanism of soybean resistance to SPB. If you are interested, we will be happy to inform you as soon as these results are published.

Point 2: The revisions to the M&M section. [M&M: The authors should describe more details for the experiential design, particularly the number of replicates for all the quantitative data. / There is a lack of a statistics section in M&M, and all the results should be described according to statistics.]

Response 2: Thank you for your suggestion . We have made the revisions according to your suggested. Please refer to sections 4.1,4.2 and 4.7 for details.

Point 3: Questions about article description and grammar. [L30: a large up-regulated genes? /L36: through integration of QTL mapping and RNA-Seq analysis - through the integration of QTL mapping and RNA-Seq analysis/L68: hundreds genes - hundreds of genes/L85: for activation - for the activation/L399: Many these genes - Many of these genes/L401: to transcript - to the transcript/L482: that as one of flavonoids - that is one of the flavonoids/L600: for each trail groups - for each trail group/ The authors need to revise grammar errors throughout the manuscript.]

Response 3: We appreciate your careful reading of this article, and we have invited native English-speaking colleague to proofread and polish the language in the revised manuscript according to your suggestion.

Thank you again for your review, and we would continue work on this subject and take full consideration with your suggestions.

Reviewer 3 Report

Soybean pod borer (SPB)-resistant and SPB-sensitive accessions (JY93 and K6) were chosen as parents for manual crossing to produce the F2 population, and a high-density linkage map was constructed using single nucleotide polymorphism (SNP) markers. A genome-scan for SPB resistant QTLs was done using the F2 population's genetic linkage map and two-year phenotypic data. They also did RNA-Seq analysis on the two parents to evaluate gene expression changes following SPB feeding in pod shells to investigate the early response mechanism of soybean to SPB. The manuscript is well structured and well discussed. However, some points should be checked and corrected before its acceptance in this journal. 

Therefore, according to my comments, I recommended the publication of the paper after major revision.

[1]   The abstract is not clear. Please add the aim and objective of the MS.

[2]   Please provide the statistical significance of the figures or tables.

[3]   Please speculate on the results. The discussion must improve.

[4]   Please provide the Conclusion section. The authors should add the significance of this research and its potential practical application.

[5]   The MS English needs to be improved. The article's English must be carefully checked for grammatical errors.

Author Response

We are grateful for your careful reading of this maunscript and your constructive comments. We have revised the manuscript in accordance with your valuable suggestions, which can be found in revised manuscript. In the meantime, we will answer each of your questions below.

Point 1: The abstract is not clear. Please add the aim and objective of the MS.

Response 1: Thank you very much for your comment. Briefly, our aim in this study was to identify the QTLs for SPB resistance in soybean and to reveal the response mechanism of soybean to SPB using transcriptome technology. These results of our works are expected to be applied in soybean breeding to address yield losses due to SPB in northeastern China. And we have revised the abstract. Thank you for reviewing it again.

Point 2: Please provide the statistical significance of the figures or tables.

Response 2: Thank you for the suggestion. In Table 1, we have added the relevant statistical analysis.

Point 3: Please speculate on the results. The discussion must improve.

Response 3: We thank you for the commons and suggestion. In accordance with your suggestions, we have revised the Results and Discussion sections.

Point 4: Please provide the Conclusion section. The authors should add the significance of this research and its potential practical application.

Response 4: Thank you for your kind suggestion. At the same time, we have revised the conclusion section.

Point 5: The MS English needs to be improved. The article's English must be carefully checked for grammatical errors.

Response 5: We appreciate your careful reading of this article, and we have invited native English-speaking colleague to proofread and polish the language in the revised manuscript according to your suggestion.

Thank you again for your through examining the manuscript and valuable suggestions.

Round 2

Reviewer 2 Report

This study screens candidate genes but still needs further validation for confirming their role. Also, the statistics are still not satisfactory to support the conclusion. Overall, the revised version did not make any significant improvement, so I have to reject it again. I encourage the authors to validate the roles of the candidate genes and re-submit.

Author Response

Response to Reviewer 2 Comments

Thank you for your constructive comments and suggestions for improving this article. We have revised the manuscript according to your valuable suggestions, which can be found in the revised manuscript.

Point 1: This study screens candidate genes but still needs further validation for confirming their role.

Response 1: Thanks for your suggestion and we also agree. Therefore, we verified the expression levels of these 18 genes in damaged pods of four additional varieties (two of them are SPB resistant and others are susceptible to SPB feeding). We further reduced the number of candidate genes from 18 to 7. The final seven candidate genes were not only derived from the results of QTL mapping and parental transcriptome, but also could have similar expression pattern in other varieties. Thus, these seven genes should play an important role in soybean resistance to SPB. Of course, the specific mechanism needs to be further verified that is also our next research content. If you are interested, we will be happy to inform you as soon as these results are published.

Point 2: Also, the statistics are still not satisfactory to support the conclusion.

Response 2: Thank you for your suggestion. Considering the effects of sample size and field environment on ESP, we still selected the t-test to detect whether or not there is a statistically significant difference in ESP between any two samples. However, the Student's t-test was performed in hypothesis testing, replacing the original paired t-test [1,2].

Thank you again for your review, and we would continue work on this subject and take full consideration with your suggestions.

References

  1. Xu, M.; Fralick, D.; Zheng, J.Z.; Wang, B.; Changyong, F. The differences and similarities between two-sample t-test and paired t-test. Shanghai archives of psychiatry2017, 29, 184.
  2. Cressie, N.A.C.; Whitford, H.J. How to Use the Two Sample t-Test. Biometrical J.1986, 28, 131-148, doi: https://doi.org/10.1002/bimj.4710280202.

Reviewer 3 Report

Requested corrections were completed.

Author Response

Response to Reviewer 3 Comments

We are very grateful for your approval of this article, and we have invited native English-speaking colleague to proofread and polish the language in the revised manuscript according to your suggestion. Thank you again for your review!

Round 3

Reviewer 2 Report

It has been improved.